# Retrospective Multicentric Study on Non-Optic CNS Tumors in Children and Adolescents with Neurofibromatosis Type 1

**DOI:** 10.3390/cancers12061426

**Published:** 2020-05-31

**Authors:** Claudia Santoro, Stefania Picariello, Federica Palladino, Pietro Spennato, Daniela Melis, Jonathan Roth, Mario Cirillo, Lucia Quaglietta, Alessandra D’Amico, Giuseppina Gaudino, Maria Chiara Meucci, Ursula Ferrara, Shlomi Constantini, Silverio Perrotta, Giuseppe Cinalli

**Affiliations:** 1Neurofibromatosis Referral Center, Department of Women’s and Children’s Health, and General and Specialized Surgery, “Luigi Vanvitelli” University of Campania, Via Luigi de Crecchio 2, 80138 Naples, Italy; stefaniapicariello34@gmail.com (S.P.); palladino.federica@gmail.com (F.P.); giusygaudino@live.it (G.G.); silverio.perrotta@unicampania.it (S.P.); 2Clinic of Child and Adolescent Neuropsychiatry, Department of Mental and Physical Health, and Preventive Medicine, “Luigi Vanvitelli” University of Campania, Largo Madonna delle Grazie 1, 80138 Naples, Italy; 3Department of Advanced Medical and Surgical Sciences, “Luigi Vanvitelli” University of Campania, P.zza L. Miraglia 2, 80138 Naples, Italy; 4Department of Pediatric Neurosurgery, Santobono-Pausilipon Children’s Hospital, Via Mario Fiore 6, 80129 Naples, Italy; pierospen@gmail.com (P.S.); mariachiarameucci@gmail.com (M.C.M.); giuseppe.cinalli@gmail.com (G.C.); 5Department of Medicine, Surgery and Dentistry, “Scuola Medica Salernitana”, Via Salvador Allende, Baronissi, 84081 Salerno, Italy; melisdaniela2418@gmail.com; 6Department of Pediatric Neurosurgery, Dana Children’s Hospital, Tel Aviv Sourasky Medical Center, 6 Weizmann St., Tel Aviv 6423906, Israel; jonaroth@gmail.com (J.R.); sconsts@netvision.net.il (S.C.); 7Department of Medicine, Surgery, Neurology, Metabolism and Geriatrics, “Luigi Vanvitelli” University of Campania, Piazza Luigi Miraglia 2, 80138 Naples, Italy; mario.cirillo@unicampania.it; 8Department of Pediatric Oncology, Santobono-Pausilipon Children’s Hospital, Via Mario Fiore 6, 80129 Naples, Italy; L.QUAGLIETTA@santobonopausilipon.it; 9Department of Advanced Biomedical Sciences, “Federico II” University of Naples, Via Sergio Pansini 5, 80100 Naples, Italy; doctoralex@hotmail.it; 10Section of Pediatrics, Department of Translational Medical Science, “Federico II” University of Naples, Via Sergio Pansini 5, 80100 Naples, Italy; charlotte-91@live.it

**Keywords:** neurofibromatosis type 1, children, CNS, brain tumors, low-grade glioma, NF1

## Abstract

The natural history of non-optic central nervous system (CNS) tumors in neurofibromatosis type 1 (NF1) is largely unknown. Here, we describe prevalence, clinical presentation, treatment, and outcome of 49 non-optic CNS tumors observed in 35 pediatric patients (0–18 years). Patient- and tumor-related data were recorded. Overall survival (OS) and progression-free survival (PFS) were evaluated. Eighteen patients (51%) harbored an optic pathway glioma (OPG) and eight (23%) had multiple non-optic CNS lesions. The majority of lesions (37/49) were managed with a wait-and-see strategy, with one regression and five reductions observed. Twenty-one lesions (42.9%) required surgical treatment. Five-year OS was 85.3%. Twenty-four patients progressed with a 5-year PFS of 41.4%. Patients with multiple low-grade gliomas progressed earlier and had a lower 5-year PFS than those with one lesion only (14.3% vs. 57.9%), irrespective of OPG co-presence. Non-optic CNS tumors are common in young patients with NF1. Neither age and symptoms at diagnosis nor tumor location influenced time to progression in our series. Patients with multiple lesions tended to have a lower age at onset and to progress earlier, but with a good OS.

## 1. Introduction

Neurofibromatosis type 1 (NF1) is a common autosomal-dominant condition with a worldwide prevalence of 1 in 3000 and an estimated incidence of 1 in 2500–3300 [1]. Neurofibromin, the protein encoded by *NF1* gene, is a negative regulator of the RAS signal transduction pathway. The haploinsufficiency of *NF1* leads to increased cell survival and proliferation mediated by hyperactivation of the Ras/PI3K signaling axis and secondary increased mTOR activation [2]. Both adults and children with NF1 are prone to developing central nervous system (CNS) tumors. Up to 15% of NF1 patients develop a brain tumor within the first two decades of life [3]. The risk of new tumor development was recently estimated to be 0.19% per year of follow-up for each NF1 patient older than 10 years [4].

CNS tumors in NF1 children are typically grade 1 pilocytic astrocytoma (PA). The most common CNS tumor is optic pathway glioma (OPG), which is also a hallmark and diagnostic criteria of NF1 [5]. A low-grade glioma (LGG) is usually indolent and is only rarely life-threatening, but may also present more aggressive behavior and be responsible for significant morbidity [6,7]. In contrast, CNS tumors in adults with NF1 tend to be of higher histological grade and have a worse prognosis than those occurring in children [8,9,10]. These observations suggest that the natural history of CNS tumors may differ depending on the age of onset. Despite the abundant body of literature on OPGs in NF1, very few studies have focused on other CNS tumors [3,4,5,8,9,10,11,12,13,14,15,16,17,18,19,20,21].

We retrospectively reviewed children and adolescents with NF1 and non-optic pathway CNS tumors followed-up and treated at two centers over a 20-year period in order to report their prevalence, clinical presentation, therapeutic approach, and outcome. We also evaluated overall survival (OS), progression-free survival (PFS), and risk factors for tumor progression.

## 2. Results

### 2.1. Study Population

Thirty-five patients were enrolled in the study. Twenty-eight patients were followed at the Neurofibromatosis Referral Center of the “Luigi Vanvitelli” University of Campania, Italy, and seven at Dana Children’s Hospital, Tel Aviv Sourasky Medical Center, Israel. Some of the patients described in the present report were included in other previously published studies [6,22,23,24]. Patient demographics and clinical data are summarized in Table 1. An MRI scan was obtained for 12 patients (34.2%) with symptoms of intracranial hypertension (4 cases); persistent headache (3 cases); neurological signs, including hemiparesis, right-side pyramidal signs, cerebellar signs, absence, and dizziness (4 cases); and endocrine disorder (growth hormone hypersecretion; 1 case). MRI was performed as a follow-up of eight patients with OPG and one with moyamoya syndrome, and to screen 13 patients for OPG and one for intellectual disability (Table 1). 

In addition, eight out of 31 patients with histological or radiological diagnosis of LGG (25.8%) had multiple metachronous lesions other than OPG: four patients had two lesions, two had three lesions, and two had four lesions (Table 2). Age at diagnosis of CNS tumor was lower in patients with multiple non-optic lesions than in patients with one lesion only (median of 7 years, interquartile range (IQR) of 5.4–10.6, range 3.2–15.8 vs. median 10.8, IQR 9.3–14.6, range 3.8–18, *p* = 0.048). There was no difference in age at diagnosis of CNS tumor in patients with or without OPG (median 9.5 years, IQR 6.4–14.3, range 3.8–16.2 vs. median 11.1 years, IQR 9.4–14.4, range 3.2–18, *p* = 0.186).

Four patients died (8.6%) at a median age of 14.25 years, IQR 8–18.56, range 6.83–19.08, at a median time from diagnosis of 0.7 years, IQR 0.11–1.81, range 0.01–2.08. Two patients with high-grade glioma (HGG) died because of tumor progression, occurring 4 months and 2.1 years from diagnosis. A third patient with intracerebral malignant peripheral nerve sheath tumor (MPNST) died as a result of post-surgical complications shortly after diagnosis, while a fourth affected by diffuse astrocytoma of corpus callosum died three years after diagnosis from another tumor, a progressive metastatic MPNST involving thoracic nerve roots from T4 to T12.

### 2.2. Tumor Characteristics and Treatment Strategies

We recorded a total of 49 tumors. Histology was available for 23 lesions (19 patients; Table 1), 18 of which (15 patients) were World Health Organization (WHO) grade 1 gliomas (14 PA and 4 gangliogliomas), one was a WHO grade 2 glioma (low-grade diffuse astrocytoma), two (2 patients) were grade 4 gliomas (glioblastoma), one was an intracranial MPNST, and one was an intracerebral schwannoma (Figure 1). The remaining 26 lesions were considered LGGs based on MRI characteristics.

Thirty-six lesions (73.4%) showed enhancement after contrast injection on T1-weighted MRI sequences, while 13 lesions did not show any enhancement in post-gadolinium MRI images.

Anatomical distribution of tumors according to histology are reported in Table 3. Most lesions (18, 36.7%) were located in the brainstem. The cerebellum was the second most common tumor location (10, 20.4%), followed by cerebral hemispheres (7, 14.3%), with other locations accounting for the remaining 28.6%. 

Seven out of 12 symptomatic patients (58.3%) had a brainstem tumor. However, brainstem lesions were more often asymptomatic (11/18). 

The initial approaches to lesions based on histology and tumor location are summarized in Table 3. A total of 25 lesions (21 patients) were treated at diagnosis or because of clinical or radiological progression. Specifically, 21 out of 49 tumors (42.9%) were treated with surgery (19 patients): six at diagnosis (12.2%), 12 during wait-and-see management (24.5%), and three after initial chemotherapy (6.1%). Thermal ablation of a PA of the right frontal lobe was performed in an 11-year-old girl.

### 2.3. Gliomas (HGGs and LGGs)

The two HGG lesions were treated with surgery, radiotherapy (60 Gy), and temozolamide (75 mg/m^2^/d). The first patient was a 6-year-old boy with HGG located in the right cerebellar hemisphere, while the second was a 17-year-old girl with HGG in the basal ganglia. The former died four months after diagnosis, while the latter 2.1 years after diagnosis, after having received cranial irradiation for an OPG 14 years previously.

Initial approaches adopted for suspected LGG lesions are listed in Table 3. Thirty-seven lesions were treated with a wait-and-see approach for a median time of 5.68 years, IQR 3.26–7.71 (range 0.14–16.27). The evolution of these patients and lesions, together with their outcome, is schematically reported in Figure 2. Specifically, 12 tumors (12 patients) required surgery after an initial observation and were located in cerebellar hemispheres (4), cerebral hemispheres (4), the basal ganglia (1), brainstem (1), corpus callosum (1), and ventricle (1). 

Five spontaneous reductions were observed after a median time of 2.58 years, IQR 1.49–4.54 (range 0.83–5.00), and one spontaneous resolution occurred 1.16 years after diagnosis (Figure 3).

One patient with OPG and multiple brain lesions developed a third PA within a lesion initially considered an unidentified bright object (UBO) of the occipital hemisphere (Figure 4).

Three patients required chemotherapy as a first-line treatment because of symptomatic, unresectable tumors [25], with a total of six lesions, three of which were located in the brainstem, two in the thalamus, and one in the right temporal lobe. Patients were treated with vincristine and carboplatin according to International Society of Pediatric Oncology (SIOP) LGG 2004 protocol [25]. Three lesions (2 patients) subsequently required surgery because of progression; a subtotal resection was achieved for a lesion of the temporal lobe, whereas lesions of the brainstem and thalamus were partially removed. All tumors remained stable following surgery.

Two lesions (2 patients) were surgically treated at diagnosis. One was a brainstem glioma causing hydrocephalus and the other was located in an anatomically critical site (cerebellar hemisphere). Both patients underwent subtotal resection (STR). The residual brainstem lesion progressed 11 months after surgery and was therefore resected again; the residual disease has remained stable for eight years. The cerebellar lesion progressed three months after the first surgery and was eventually completely resected, with no further relapse for three years.

A total of 16 LGGs were treated by surgery, with a curative gross total resection (GTR) performed in eight of them. Of the remaining eight lesions that were not completely removed at the first attempt, three required further surgery and five remained stable.

### 2.4. Schwannoma and MPNST

The intracerebral parietal schwannoma was asymptomatic and diagnosed by screening MRI (Figure 1). The lesion was completely removed and did not show any signs of relapse during a follow-up of 6.75 years.

The MPNST, arising from the brainstem and occupying the occipital foramen, was surgically treated at diagnosis because of intracranial hypertension and severe brainstem compression. The patient died following a stormy post-operative period complicated by hydrocephalus, swallowing problems, tracheostomy, and cerebrospinal fluid mycotic infection. 

### 2.5. Survival Analyses and Predictive Factors

Whole population 5-year OS was 85.3%, irrespective of tumor histology and treatment (Figure 5). Twenty-four patients out of 35 progressed during follow-up with a 5-year PFS of 41.4% (Figure 5). No significant difference was found in median time to progression for the following binary variables: gender (*p* = 0.24), age at diagnosis less or more than 10 years (*p* = 0.92), tumor location (posterior fossa vs. others, *p* = 0.77; brainstem vs. others, *p* = 0.60; midline supratentorial vs. others, *p* = 0.84).

When considering LGGs, patients with multiple lesions (outside the optic pathway) progressed earlier than those with one lesion only, with a median time to progression of 1.52 years (95% confidence interval (CI) 0.11–2.94) vs. 5.76 years (95% CI 2.5–9.0), respectively (*p* = 0.03). Five-year PFS was 14.3% for patients with multiple non-optic lesions and 57.9% for patients with one lesion only (Figure 6). 

No significant difference was found in median time to progression in patients with and without OPG (4.99 years (95% CI 0–10.23) vs. 3.66 years (95% CI 0–7.37), respectively, *p* = 0.63). A similar 5-year PFS was found in both groups (42.8% for patients with OPG vs. 48.2% for patients without OPG). None of the LGG patients received chemotherapy for OPG during the observation time.

The presence of multiple non-optic LGG lesions proved to be the only independent predictor of progression in the univariate analysis (Hazard Ratio 2.81 (CI 95% 1.06–7.46), *p* = 0.038), while the other explored variables were not found statistically significant (Appendix A).

### 2.6. Genetic Findings

*NF1* mutations were identified in 18 patients, of which 17 carried truncating mutations. The mutation type and effect together with the inheritance pattern are reported in Table 4 for each patient included in the study.

## 3. Discussion

To the best of our knowledge, our series represents the largest cohort of NF1 children and adolescents affected by extra-optic CNS tumors [4,5,8,12,26,27,28,29,30,31]. In our series, almost half of lesions were incidentally detected (23/49, 46.9%). MRI was requested mainly (13) for OPG screening. Although the efficacy of routine brain MRI screening is still debated in NF1, clinicians should be aware of the opportunity to detect and manage an incidental brain tumor in this population. However, a negative scan does not exclude the possibility that a patient may later develop a brain tumor.

In our cohort, overall prognosis was good, with a 5-year OS of 85.3%, in line with previous reports [8]. Nevertheless, despite an excellent survival rate, 5-year PFS was 41.4% (Figure 5). Contrary to the findings of previous studies, neither age and symptoms at diagnosis nor tumor location influenced progression timing in our series [17,32,33,34].

In terms of the location of lesions, the posterior fossa was confirmed to be the most commonly affected site, followed by the brainstem (18, 36.7%) and cerebellum (10, 20.4%) [17,26,27,28,29] (Table 3).

Our population included a large number of LGGs (91.8% of all lesions). LGGs are in fact the most common CNS tumors in NF1, with PA representing the most prevalent histological subtype [5,30]. We also recorded rarer histologies in our population, including HGGs, MPNST, and schwannoma.

We reported a prevalence of 26% of metachronous extra-optic LGGs in our pediatric and adolescent population with NF1 (8/31). In 2017, Sellmer et al. found 20% of patients with multiple non-optic gliomas in their series, including both adults and children [4]. In 2003, Guillamo et al. reported the same percentage of multiple CNS tumors, but they included OPGs [8]. Furthermore, they found that patients with multiple lesions were not affected by higher mortality. In agreement, none of our patients with metachronous lesions died. 

In our study, patients with multiple lesions had a greater risk of progression and a lower 5-year PFS than patients with a single lesion, whereas the co-presence of OPG was not a risk factor for progression (Appendix A). Although nearly all multifocal lesions radiologically worsened, patients tended to remain asymptomatic. These findings support the evidence that multiple CNS tumors, especially LGGs, are a hallmark of NF1 [8,31,35], and even in the case of multiple lesions and radiological progressions, patients could still be asymptomatic [7,34]. Tumor molecular biology may be different in this category of patients. It was in fact speculated that a patient is more likely to develop additional non-optic tumors if they have already developed one [4]. Our patients with multiple non-optic lesions have a lower CNS tumor onset age than those with a single non-optic lesion, strengthening this hypothesis. 

Here, 13 out of 45 LGG lesions (28.9%) did not show any enhancement on post-contrast MRI images. LGG in NF1 children might be enhanced heterogeneously [36] or not at all [37,38,39]. This observation makes the differentiation between LGGs and UBOs even more challenging. UBOs are non-enhancing T2 hyperintensities typical of NF1 found in the brain and medulla [39,40], which are described as potential precursors of brain tumors [41]. In agreement, a boy with OPG and multiple lesions in our population developed a third PA within a lesion initially considered as a UBO of the occipital hemisphere (Figure 4). The absence of contrast enhancement could, therefore, be misleading in diagnosis of LGG [37].

In terms of therapy, 60% of patients in our series received treatment for a CNS tumor. In the case of LGG, chemotherapy was the treatment option for symptomatic children with unresectable lesions [25], while GTR was confirmed curative in eight out of 16 tumors treated by surgery (50%), demonstrating its key role in managing LGG [25,42]. One patient previously treated for OPG in the first year of life and for aqueductal stenosis at the age of 7 developed a PA of the right frontal lobe at the age of 11 and underwent thermal ablation (LITT procedure) of this lesion. Although the long-term effects of laser ablation in patients with NF1 are not yet known, this technique is more precise, less invasive, does not involve ionizing radiations in terms of volume of tissue ablation compared to other methods, and allows complete resection, avoiding collateral damage [43,44,45,46].

A wait-and-see approach was adopted as a first-line strategy for almost 75% of patients and lesions in our series. Although one-third of tumors required further surgery, 16% did not progress, 13% showed radiological reduction, and one disappeared (Figure 3). Cases of spontaneous resolution of brain gliomas in NF1 patients are described in the literature [47,48,49,50,51]. Therefore, when managing an asymptomatic lesion without radiological signs of high-grade histology, a wait-and-see policy should be taken [25]. 

Although NF1-associated gliomas are usually PA, patients with NF1 may present high-grade gliomas, especially in adulthood, with a poor prognosis [52,53,54]. In our series, the two patients with glioblastoma both died, despite surgery and adjuvant chemoradiotherapy [55]. One developed a basal ganglia glioblastoma 14 years after cranial irradiation for OPG. In the past, radiotherapy has been used to treat OPGs [56,57,58], achieving a high rate of tumor control [56,57,59]. It is now avoided in NF1-related LGGs, due to the known risk of secondary malignancies and radiation-induced vasculopathies [25,58,60]. We also recorded the case of an 11-year-old patient with intracerebral MPNST and a 4-year-old with schwannoma (Figure 1), which are both very rare in NF1 patients and exceptionally rare in childhood [61,62,63,64,65,66,67].

From the genetic perspective, the number of genotype–phenotype associations in NF1 continues to increase. NF1 gene has one of the highest spontaneous mutation rates and more than 3000 mutations have been recognized to date [68]. The following variants are associated with specific phenotypes: missense variants at the *NF1* codons p.Arg1809, p.Met1149, p.Arg1276, and p.Lys1423 and the in-frame deletion c.2970-2972 delAAT [69,70,71]. In our series, 19 out of 35 patients underwent molecular testing for NF1 and an *NF1* mutation was identified in 18 cases (Table 4). Seventeen of these mutations were predicted to produce a truncated neurofibromin.

*NF1* germline mutations were shown to result in different levels of neurofibromin expression in NF1 patient fibroblasts, ranging from 25% to 75% [72]. This finding supports the hypothesis that not all *NF1* mutations are equivalent and that residual amounts of functionally active neurofibromin might be linked to the phenotype [73]. Genotype–phenotype associations and the parent-of-origin effect overall or by patient sex were investigated and subsequently excluded for OPGs in NF1 [74,75,76]. It might be interesting to extend such studies and test this hypothesis in brain tumors. Any attempt to associate observed genotypes and brain tumors in our cohort of patients was limited by the very low number of cases examined. Although somatic *NF1* gene inactivation is required for NF1-related tumorigenesis [77], we could speculate that: (1) the observed higher rate of truncating mutations in *NF1* may suggest that at least 50% loss-of-function is necessary to initiate tumorigenesis; (2) additional genetic modifiers unlinked to the *NF1* locus might play a role in brain tumors, as we proposed for NF1-related moyamoya vasculopathy [78].

Major limitations of the present study are its retrospective design and the lack of a molecular tumor profile. Our molecular and immunological understanding of NF1-associated gliomas is rapidly evolving, and biopsy is increasingly indicated not only for histologic confirmation, but also for molecular-targeted therapy [42]. In terms of new therapeutic options, The MEK1/2 inhibitor selumetinib seems to be active in NF1-related pediatric LGG [79]. The efficacy of selumitinib versus standard chemotherapy in both newly diagnosed and progressive or recurrent tumors, irrespective of biopsy, is currently under evaluation in a phase III trial (clinical trial.gov number NCT03871257). 

## 4. Materials and Methods

This retrospective multicentric study involved patients (0–19 years of age) affected by NF1, according to NIH NF1 criteria (1988) [80] and diagnosed with non-optic CNS tumors from January 2000 to January 2019 at two centers. 

Brain MRI images of all enrolled patients were reviewed. Any measurable area in at least two dimensions of high signal intensity on T2-weighted images was considered a tumor if it had one of the following characteristics: gadolinium enhancement, mass effect, peripheral edema, or a mural nodule associated with a cystic or necrotic component [8,38,39].

Histological confirmation of tumors was not mandatory for patient inclusion. Specifically, lesions with typical characteristics of LGGs did not undergo biopsy [7,37,42]. In contrast, histological examination was performed for lesions with radiological features of high-grade histology (high degree of tumor heterogeneity and contrast enhancement, restricted diffusion on diffusion-weighted MRI, and increased relative cerebral blood volume on perfusion-weighted MRI) [36,38,42]. Tumors with aggressive behavior and those requiring surgery were also subjected to histological examination.

Collected data included demographic characteristics (gender, age at diagnosis, and inheritance of NF1), age at diagnosis of CNS tumors and NF1, and clinical signs or symptoms at diagnosis of CNS tumors. Indications for brain MRI scan at diagnosis were also recorded. When a patient presented multiple lesions, data on all lesions were collected. We documented the anatomical site, enhancement of all lesions, and histology when available [81], as well as the co-presence of OPG. We also recorded the management of tumors, including wait-and-see, surgery, chemotherapy, and radiotherapy approaches. We collected data on the type of surgical resection (gross total- and subtotal- resection (respectively GTR and STR) [82,83], duration of follow-up, recurrence, progression, or resolution. Ethical approval from local committees was obtained. 

### 4.1. Statistical Analysis

Continuous non-parametric variables are presented as the median, IQR, and range, whereas categorical variables are expressed as number and percentage. Mann–Whitney U test was used to compare continuous non-parametric variables. Kaplan–Meier analysis was run to determine OS and PFS [84]. OS was calculated from date of diagnosis until death from any cause. PFS was measured from date of diagnosis until radiological or clinical progression date. Specifically, for multiple lesions, the first radiological progression of one lesion was considered as the progression date, irrespective of the evolution of any others. For both OS and PFS analyses, patients were censored at last available follow-up time if no event occurred. 

Factors that may influence time to progression (gender, age at tumor diagnosis, location of tumor, symptoms at diagnosis) were tested by comparing PFS curves with the log-rank test.

Exclusively for patients with histological or radiological diagnosis of LGG, log-rank test was used to compare PFS curves of patients with and without associated OPG, and those of children with single and multiple lesions. The Cox regression model was used to explore predictors of progression in patients with LGG. For all analyses, *p* values < 0.05 were considered statistically significant. IBM SPSS Statistics 22 Software for Windows was used for statistical analysis.

### 4.2. Genetic Findings

*NF1* mutations were recorded in terms of the genomic and protein location and type of mutation. *NF1* germline mutations were obtained from a review of clinical records and no further genetic analyses were performed during the study.

## 5. Conclusions

Extra-OPG CNS tumors are a relatively common malignancy in children and adolescents with NF1. Although other histological types do rarely occur, LGGs are the most common CNS tumors in this population. In our experience, LGG lesions tend to be and remain asymptomatic in young patients, regardless of radiological progression. In addition, patients with multiple metachronous LGGs have a lower age at onset and tend to progress earlier than patients with one lesion only, irrespective of the co-presence of an OPG. Hence, in line with the consensus very recently published by Packer et al. in 2020 [42], our experience confirms that a wait-and-see approach is initially advised for asymptomatic lesions with radiological characteristics of low-grade tumors. Conversely, surgery remains the best therapeutic option in symptomatic LGGs, with the aim of achieving complete resection.

## Figures and Tables

**Figure 1 cancers-12-01426-f001:**
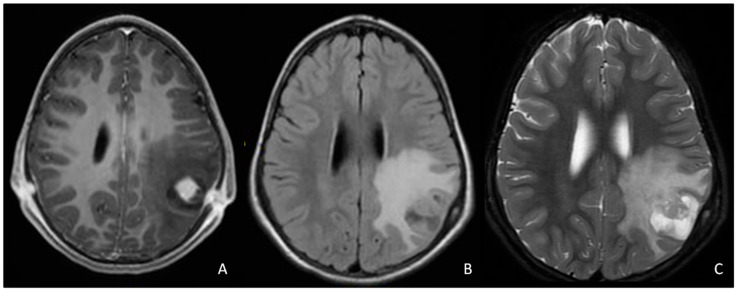
MRI images of the left parietal intracerebral schwannoma: (**A**) contrasted enhanced T1-weighted image; (**B**) FLAIR; (**C**) T2-weighted image.

**Figure 2 cancers-12-01426-f002:**
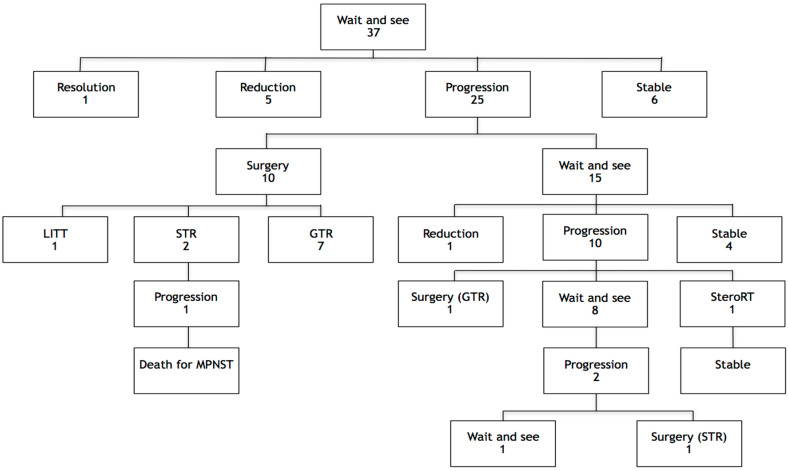
Evolution of the 37 lesions initially managed with a wait-and-see strategy, together with further approaches and outcome. LITT, laser interstitial thermal therapy; GTR, gross total resection; STR, subtotal resection; MPNST, malignant peripheral nerve sheath tumor; StereoRT, stereotactic radiotherapy.

**Figure 3 cancers-12-01426-f003:**
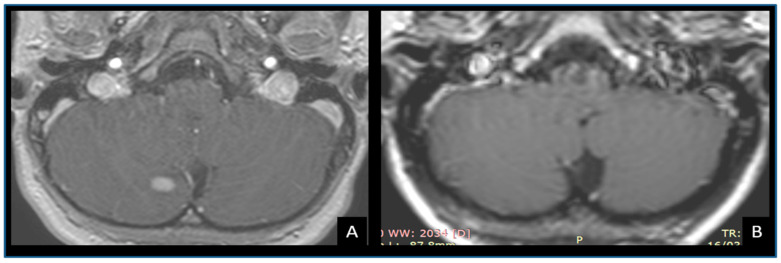
MRI images of an enhancing cerebellar lesion located in the right paravermian region (**A**). The lesion was radiologically followed every 6 months and was undetectable at 14 months of radiological follow-up (**B**).

**Figure 4 cancers-12-01426-f004:**
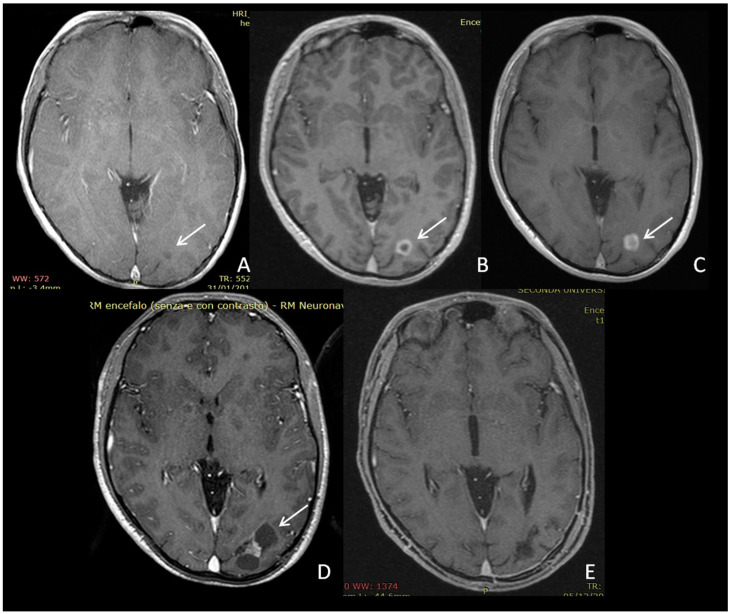
MRI images (contrast enhanced T1-weighted) demonstrating evolution of an occipital unidentified bright object (UBO) in an expansive lesion in a 9-year-old boy. The lesion appeared as a subcentimetric hypointense non-enhancing area in 2011 (**A**, arrow). In 2013, the hypointense lesion was surrounded by a ring of contrast enhancement (**B**). In 2014, the contrast enhancement appearance was more diffuse (**C**). Six months later, a mixed cystic–solid tumor was detected (**D**). Post-operative image (**E**). The tumor histology was confirmed as a pilocytic astrocytoma.

**Figure 5 cancers-12-01426-f005:**
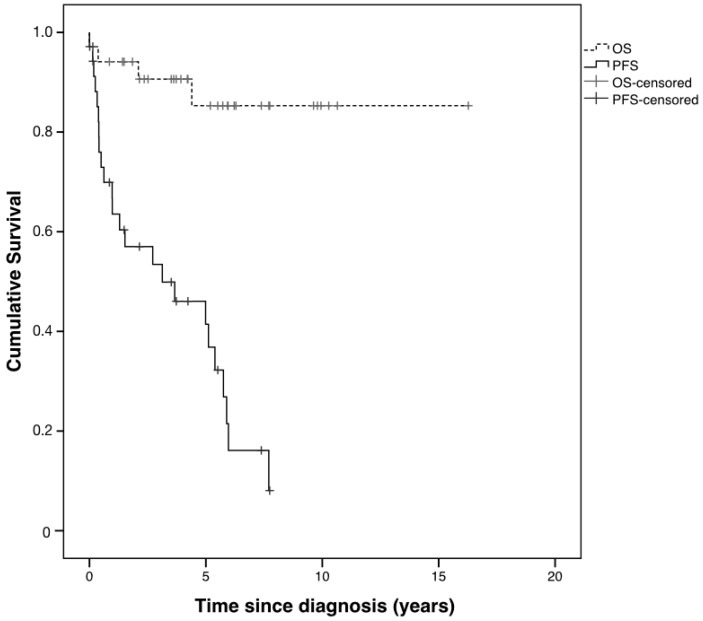
Kaplan–Meier whole population overall survival (OS) and progression-free (PS) survival curves.

**Figure 6 cancers-12-01426-f006:**
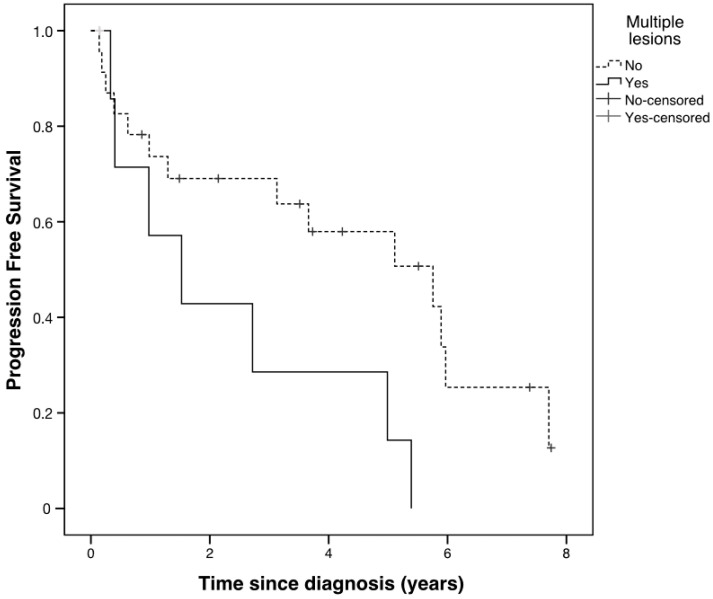
Kaplan–Meier progression-free survival (PFS) curves by non-optic multiple lesions in patients with low-grade gliomas. Please note that just one patient was censured among those with multiple lesions and the relative hyphen is in light grey.

**Table 1 cancers-12-01426-t001:** Patient demographics and characteristics. Categorical variables are expressed as a number and percentage. Continuous non-parametric variables are reported as the median, interquartile range (IQR), and range. NF1, neurofibromatosis type 1, CNS; central nervous system, MRI; magnetic resonance imaging; OPG, optic pathway glioma; MPNST, malignant peripheral nerve sheath tumor.

Demographic and Clinical Characteristics	Study Population 35 Patients
Gender (male)	15 (42.9%)
NF1 inheritance	
● Sporadic	19 (54.3%)
● Maternal	10 (28.5%)
● Paternal	4 (11.5%)
● Unknown	1 (2.8%)
Age at diagnosis of NF1 (years)	2 [0.5–7] (0–15.25)
Age at diagnosis of CNS tumor (years)	10.4 [7.3–14.3] (3.2–18)
Follow-up duration (years)	4.2 [2.1–7.4] (0.01–16.3)
Indications for MRI scan	
● Screening	13 (37.1%)
● Follow-up of OPG	8 (22.8%)
● Follow-up of moyamoya	1 (2.9%)
● Intellectual disability	1 (2.9%)
● Unspecified headache	3 (8.6%)
● Intracranial hypertension	4 (11.4%)
● Other neurological signs	4 (11.4%)
● Endocrine disorders	1 (2.9%)
Tumor histology	
● Low-grade glioma	15 (42.8%)
● High-grade glioma	2 (5.7%)
● MPNST	1 (2.9%)
● Schwannoma	1 (2.9%)
● Not available	16 (45.7%)
Patients with concomitant OPGs	18 (51%)
Number of non-optic lesions per patient	
● 1	27 (77.2%)
● 2	4 (11.4%)
● 3	2 (5.7%)
● 4	2 (5.7%)
Deaths	4 (11.4%)

**Table 2 cancers-12-01426-t002:** Tumor-related characteristics and treatments of the eight patients with multiple non-optic tumors. GTR: gross total resection; PA, pilocytic astrocytoma; LITT: laser interstitial thermal therapy; STR: subtotal resection.

Patient ID	OPG	Number of Lesions	Symptoms at Diagnosis	Lesion Locations	Treatment/Biopsy	Histology
1	Yes	2	Headache	⮚ Brainstem (midbrain) ⮚ Thalamus	⮚ Chemotherapy ⮚ GTR of both lesions	Ganglioglioma
2	Yes	2	Hemiparesis	⮚ Brainstem (midbrain) ⮚ Right temporal lobe	⮚ Chemotherapy ⮚ GTR of the right temporal lobe lesion ⮚ Biopsy of the brainstem lesion	PA
3	Yes	2	Intracranial hypertension	⮚ Brainstem (midbrain) ⮚ Thalamus	⮚ Chemotherapy ⮚ Ventriculo-Peritoneal shunt	-
4	Yes	2	-	⮚ Brainstem (midbrain) ⮚ Right frontal lobe	⮚ LITT of the right frontal lobe lesion	PA
5	Yes	3	-	⮚ Brainstem (midbrain) ⮚ Corpus callosum ⮚ Left occipital lobe	⮚ GTR of the left occipital lobe lesion	PA
6	No	3	-	⮚ Right Cerebellar Hemisphere ⮚ Left paravermian ⮚ Left Cerebellar Hemisphere	⮚ STR and subsequent GTR of the right cerebellar hemisphere lesion ⮚ GTR of the left paravermian lesion	PA
7	Yes	4	Ataxia and dysphagia	⮚ Right cerebellar hemisphere ⮚ Left fornix ⮚ Brainstem (pons) ⮚ Brainstem (medulla)	⮚ GTR of the right cerebellar hemisphere lesion	PA
8	Yes	4	-	⮚ Corpus callosum ⮚ Right cerebral hemisphere (rolandic) ⮚ Left cerebellar hemisphere ⮚ Brainstem (midbrain)	⮚ GTR of the right cerebral hemisphere lesion	PA

**Table 3 cancers-12-01426-t003:** The initial approach to lesions based on histology and tumor location. CT, chemotherapy; RT, radiotherapy; HGGs high grade gliomas; MPNST, malignant peripheral sheet tumor; LGGs, low grade gliomas.

Histology/Location	All Lesions (*n* = 49)	Wait-and-See (*n* = 37)	Surgery (*n* = 5)	CT (*n* = 6)	Surgery + CT + RT (*n* = 2)
HGGs					
Basal ganglia	1	-	-	-	1
Cerebellum	1	-	-	-	1
MPNST					
Brainstem	1	-	1	-	-
Schwannoma					
Parietal	1	-	1	-	-
LGGs					
Brainstem	17	13	1	3	-
Cerebellum	9	8	1	-	-
Cerebral lobe	6	5	-	1	-
Basal ganglia	2	2	-	-	-
Corpus callosum	3	3	-	-	-
Ventricles	3	3	-	-	-
Hypothalamus	2	2	-	-	-
Thalamus	2	-	-	2	-
Fornix	1	1	-	-	-

**Table 4 cancers-12-01426-t004:** NF1 molecular findings of all included patients. *NF1* reference sequence: NM_000267.3.

ID	DNA Change	Protein Change	Effect Type	Inheritance
S01	492_502del	Cys167Glnfs*10	Frame-shift	Sporadic
S02	-	-	-	Maternal
S03	574C>T	Arg192*	Nonsense	Maternal
S04	1642-?_4772+?del	Ala548Valfs*9	Intragenic deletion (exons 15–36)	Sporadic
S05	2851G>T	Leu952Cysfs*22	Splicing	Sporadic
S06	5839C>T	Arg1947*	Nonsense	Maternal
S07	-	-	-	Paternal
S08	1260+1604A>G	Ser421Leufs*4	Intronic cryptic splice site	Sporadic
S09	-	-	-	Sporadic
S10	-	-	-	Sporadic
S11	-	-	-	Paternal
S12	1863del	Cys622Valfs*9	Frame-shift	Paternal
S13	1466A>G	Tyr489*	Cryptic splice site	Sporadic
S14	-	-	-	Sporadic
S15	-	-	-	Sporadic
S16	5592_5596del	Asn1864Lysfs*26	Truncating	Maternal
S17	4840_4854del	Tyr1614_Tyr1618del	In-frame deletion	Sporadic
S18	5264C>G	Ser1755*	Nonsense	Maternal
S19	-	-	-	Sporadic
F01	analyzed	mutation not found RT PCR and mlpa		Sporadic
F02	-	-	-	Maternal
F03	1863del	Cys622Valfs*9	Frame-shift	
F04	5928G>A	Trp1976*	Nonsense	Sporadic
F05	-	-	-	Maternal
F06	2446C>T	Arg816*	Nonsense	Sporadic
F07	7778del	Lys2593Argfs*10	Frame-shift	Sporadic
P01	2041C>T	Arg681*	Nonsense	Paternal
P02	-	-	-	Sporadic
P03	-	-	-	Sporadic
I01	4246dup	Arg1416Lysfs*30	Frame-shift	Sporadic
I02	3415_3416del	Ala1139Ilefs*55	Frame-shift	Maternal
I03	-	-	-	Maternal
I04	-	-	-	Sporadic
I05	-	-	-	n.a.
I06	-	-	-	Maternal

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
