# Peer review of "Retrospective Multicentric Study on Non-Optic CNS Tumors in Children and Adolescents with Neurofibromatosis Type 1"

_cancers, 2020, doi:10.3390/cancers12061426_

Round 1
Reviewer 1 Report
This is a very interesting paper & methodology is thoroughly carried out & well documented. I would have liked to have seen more reference to genetics. I note that all patients met NF1 diagnostic criteria but it would be useful to know their NF1 gene mutations. If this was incomplete it is still worth mentioning in the discussion. If patients were genotyped, this should be included, in Table 1 & the discussion, to improve understanding of genotype phenotype correlation & its potential usefulness as a prognostic indicator. It would also be useful to confirm an NF1 gene mutation in the child with a schwannoma to rule out schwannomatosis.
Author Response
Dear Reviewer 3,
Thank you very much for the interesting suggestion.
Given the retrospective nature of the paper we did have few genetic data available that we have reported in a further table, table 4, devoted to NF1 molecular status.
We also added to the revised version of our manuscript: the molecular data recording methods, a short paragraph into the results and a paragraph dedicated to interpretation of data respect of literature and actual knowledge into the discussion.
Regarding , in detail, the patient with intracranial schwannoma he did not receive any molecular testing yet inherited NF1 from his mother (who also inherited the NF1, in this case from her father), hence there is no doubt it is a classical NF1.
We thank you and the other reviewers very much for the suggestions . The manuscript has been carefully checked and appropriate changes have been made in accordance with all your comments. We hope that the revised manuscript is now suitable for publication.
Reviewer 2 Report
Santoro et al present a large natural history study of non-optic central nervous system tumors in 31 NF1 subjects. They describe the demographic, surgical and medical treatments, and include the natural history including time to progression. They show that patients with multiple brain lesions had a lower age at onset and progressed earlier. LGGs were the most common tumors. A few suggestions for improvement:
- OPGs have been associated with growth hormone excess and gigantism in NF1. Please comment on whether there was a growth spurt in your pediatric population, and if so, provide appropriate growth charts. It would be useful to know their IGF-1 levels/z-score, etc
- It would also be useful to look at their pituitary glands via existing imaging and preferably MRIs to see if voluminous and suggestive of hyperplasia, as somatommamotroph hyperplasia has been implicated in growth hormone excess in NF1
- For table 1, inheritance does not add up to total of 35 patients. If data is unkown, please indicate as a footnote. Please ensure that all variables add up.
- Here are some refs to aid with the above suggestions:
https://www.ncbi.nlm.nih.gov/pubmed/30283094
https://www.ncbi.nlm.nih.gov/pubmed/29070623
https://www.ncbi.nlm.nih.gov/pubmed/28631895
https://www.ncbi.nlm.nih.gov/pubmed/27348432
Author Response
Dear Reviewer 2,
Thank you very much for all your suggestions.
The manuscript has been carefully checked and appropriate changes have been made in accordance with your comments, below a point-by-point response.
1. Thank you very much for your suggestion. We agree that this is a very interesting topic yet we sadly did not focus on OPGs. We aimed to report on the non-optic brain tumors in NF1. Hence we did not record growth velocity neither IGF1 levels in our population. OPG co-presence was only recorded to look for any association between OPG and non-optic tumors. Just one patient presented with growth hormone excess that led to diagnose an hypothalamic LGG.
2. Thank you for your suggestion, yet we did not focus on this aspect. The only one patient with growth hormone excess did have a hypothalamic LGG and pituitary gland was normal on MRI scan.
3. Yes, it is right. We amended data adding paternally inherited forms in table 1.
An additional table (table 4) reporting genetic findings and inheritance of NF1 for all included patients has also been added to the revised manuscript.
Finally the manuscript has been edited by a native English speaker and all made corrections are tracked.
We thank you and the other reviewers very much for the suggestions . The manuscript has been carefully checked and appropriate changes have been made in accordance with all your comments. We hope that the revised manuscript is now suitable for publication.
Reviewer 3 Report
Summary of the Key Findings of the Study:
The NF1 signaling pathway has a wide range of influences on normal cell physiology, and its deregulation plays a significant role on non-optic CNS tumors in children. This retrospective multicentric study contributes to better delineate the clinical features associated with patient’s disease course.
General Critique of Work
Minor issues to be addressed:
- i) Manuscript layout and word are partially respected:
-Author names are spelled out and institutional affiliations are signified with footnotes. Likewise, corresponding authors are noted with an asterisk in the author list. Each author is well cited in the Author’s contributions part. However, police size in the author’s affiliation seems different for the reference f with a potential mistake in “P.zza L. Miraglia”.
The MD title could be removed decreasing thus the size of author’s names.
-By following journal recommendations, authors have to add three or ten pertinent keywords. In the present case, 7 keywords were counted without for some of them a real pertinence. Authors should increase or decrease the number of key word.
-OPG for “optic pathway glioma” is not defined in the abstract which count only 190 words.
-Line 128 “temozolamide”
-STR and GTR are defined in the figure 2 while they appear in early in the manuscript
-Line 146 in the legend of Figure 3 “radiologi-cal”
-Authors have to define earlier UBO term
Major issues to be addressed:
1-Authors have performed Kaplan Meier analysis while hazard ratio remains missing. That will be useful to obtain these results in a manner to obtain the risk.
2-Authors provide several clinical features while they didn’t perform multiple regression analyses to improve their conclusion. The patient’s number remains perhaps lower to perform such analysis. Whatever authors have to discuss these crucial points.
3-It will be interesting to see at least three patients for their NF1 status by sequencing with methodology.
Conclusion:
Honestly; authors do a fairly good job of explaining the potentially high impact of their findings. However, as noted in several points above, there are some unclear terms not explained such STR, GTR. NF1 sequencing remains missing while the study remains based on this clinical feature. 1) Reduce the number of the patient might impact statistical analysis. Why didn’t used cox regression or multiple regression analyses. These suggestions would improve the manuscript.
Author Response
Dear Reviewer 3,
Thank you very much for all your suggestions.
The manuscript has been carefully checked and appropriate changes have been made in accordance with the your comments, below a point-by-point response.
Minor revisions
1.-Author names are spelled out and institutional affiliations are signified with footnotes. Likewise, corresponding authors are noted with an asterisk in the author list. Each author is well cited in the Author’s contributions part. However, police size in the author’s affiliation seems different for the reference f with a potential mistake in “P.zza L. Miraglia”.
We thank the reviewer for the thorough revision.
We checked all data regarding affiliations and they are right. Each department, even if belonging to the same university, has got a specific address.
The MD title could be removed decreasing thus the size of author’s names.
We apologize for making this mistake, appropriate corrections were made
-By following journal recommendations, authors have to add three or ten pertinent keywords. In the present case, 7 keywords were counted without for some of them a real pertinence. Authors should increase or decrease the number of key word.
We modified the keywords following reviewer suggestion
-OPG for “optic pathway glioma” is not defined in the abstract which count only 190 words.
We added it following the reviewer’s suggestion
-Line 128 “temozolamide”
Sorry. It was a typo. We have corrected the spelling
-STR and GTR are defined in the figure 2 while they appear in early in the manuscript
Sorry for the mistake. We defined GRT, SRT and LITT abbreviations in the figure 2, the first point where they appeared.
-Line 146 in the legend of Figure 3 “radiologi-cal”
We amended the text
-Authors have to define earlier UBO term
Definition of UBOs was correctly reported in the manuscript.
Major issues to be addressed:
1-Authors have performed Kaplan Meier analysis while hazard ratio remains missing. That will be useful to obtain these results in a manner to obtain the risk.
We thank the reviewer for the suggestion. We have added univariate Cox regression result with Hazard Ratio in the manuscript.
2-Authors provide several clinical features while they didn’t perform multiple regression analyses to improve their conclusion. The patient’s number remains perhaps lower to perform such analysis. Whatever authors have to discuss these crucial points.
We thank the reviewer for the suggestion.
As pointed out, the number of patients and the number of events for each variables was low to perform a powerful multivariate analysis. Furthermore, the univariate analyses did not show any significant predictor to include in the multivariate analysis, apart from the presence of multiple lesions.
However, we have added in the methods that we performed the analysis, in the results the following sentence: “Presence of multiple non-optic LGG lesions represented the only independent predictor of progression in the univariate analysis (Hazard Ratio 2.81 (C.I. 95% 1.06 – 7.46), p 0.038), while other explored variables were not found significant (supplementary table 1).” While in the discussion ewe reported”.
We added a supplementary table titled “Univariate Cox regression analyses of PFS in patients with LGG “
3-It will be interesting to see at least three patients for their NF1 status by sequencing with methodology.
We thank the reviewer for this useful suggestion. An additional table reporting genetic findings and inheritance of NF1 of included patients has been added to the revised manuscript.
Conclusion:
Honestly; authors do a fairly good job of explaining the potentially high impact of their findings. However, as noted in several points above, there are some unclear terms not explained such STR, GTR. NF1 sequencing remains missing while the study remains based on this clinical feature. 1 Reduce the number of the patient might impact statistical analysis. Why didn’t used cox regression or multiple regression analyses. These suggestions would improve the manuscript
We thank you and the other reviewers very much for the suggestions . The manuscript has been carefully checked and appropriate changes have been made in accordance with all your comments. We hope that the revised manuscript is now suitable for publication.
Reviewer 4 Report
The authors present a retrospective series of CNS tumors (excluding optic gliomas) in pediatric patients with neurofibromatosis I, who were followed up in their institution
There are no major problems with the design of the study and the data analysis and the contribution to the knowledge of those tumors is interesting. However, the study shows all the limits of the retrospective series, as follows:
- About half of the patients were symptomatic and half have been detected during screening. These are two completely different categories of patients, since many of the latter would probably have never been diagnosed, in the absence of any symptom
- The analysis of the data is very detailed. The authors could erase figure 6 and 7 (few data) and try to make some comparison between the symptomatic and asymptomatic tumors.
- Any biological characterization is missing: the authors acknowledge this fact, but anyway we remain at the starting point with two different categories of tumors and we do not know why. In this respect the study lacks novelty
- Discussion: the authors could make some consideration about the issue of brain MRI screening in the light of their results
- Syntax and English language: The paper is difficult to read. The authors should take a better care of the writing, refraining from one-line sentences which are often not related one to another (especially Intro and Discussion)
Author Response
Dear Reviewer 4,
Thank you very much for all your suggestions.
The manuscript has been carefully checked and appropriate changes have been made in accordance with your comments, below a point-by-point response.
About half of the patients were symptomatic and half have been detected during screening. These are two completely different categories of patients, since many of the latter would probably have never been diagnosed, in the absence of any symptom
We thank the reviewer for the suggestion. We report 12/35 symptomatic patients. Two of them had histologies other than LGG (MPNST and High Grade Glioma). In detail, 10 patients out of 31 presenting with LGGs were symptomatic at diagnosis.
We used Kaplan-Meier Method and Log-rank test to compare progression timing between patients with and without symptoms at diagnosis, for both the whole population and the LGG subgroup, but this difference was not significant:
- LGG-patients, symptomatic vs asymptomatic: 5.39 years (95% C.I 2.50 – 8.38) vs 2.72 years (95% C.I 0.0 – 6.35) , p 0.122.
- Whole population symptomatic vs asymptomatic: 4.99 years (95% C.I 0.00 – 10.13 ) vs 2.72 years (95% C.I 0.00 – 6.00) respectively, p 0.275.
The analysis of the data is very detailed. The authors could erase figure 6 and 7 (few data) and try to make some comparison between the symptomatic and asymptomatic tumors.
We thank the reviewer for his/her comment. As reported above, we did not find any significant difference in PFS between symptomatic and asymptomatic patients. On the other hand, patients with multiple non-optic LGG lesions progressed earlier than those with one lesion only, with a median time to progression of 1.52 years (95%CI 0.11 – 2.94) vs 5.76 years (95% CI 2.5 – 9.0) respectively (p 0.03). Five-year-PFS was 14.3 % for patients with multiple non-optic lesions and 57.9 % in patients with one lesion only. Furthermore, following reviewer 3 suggestion we performed Cox regression analysis in order to obtain Hazard Ratio. Presence of multiple non-optic LGG lesions represented the only independent predictor of progression in the univariate analysis (HR 2.81 (C.I. 95% 1.06 – 7.46), p 0.038), while other explored variables were not found significant. For these reasons, if the reviewer and the editor agree, we would like to keep figure 6, while we could delete figure 7.
Any biological characterization is missing: the authors acknowledge this fact, but anyway we remain at the starting point with two different categories of tumors and we do not know why. In this respect the study lacks novelty
We perfectly agree with the reviewer and in this sense future researches are needed to verify if any biological different signature is associated to different tumour behaviour.
Discussion: the authors could make some consideration about the issue of brain MRI screening in the light of their results
We thank the reviewer for raising this interesting question. We added a sentence in the discussion section debating this chance. “
"Although the efficacy of routine brain MRI screening is still debated in NF1, clinicians should be aware of the opportunity to detect and manage an incidental brain tumor in this population. However, a negative scan does not exclude the possibility that a patient may later develop a brain tumor. “
Syntax and English language: The paper is difficult to read. The authors should take a better care of the writing, refraining from one-line sentences which are often not related one to another (especially Intro and Discussion)
The manuscript has been edited by a native English speaker and all made corrections are tracked.
Round 2
Reviewer 3 Report
Dear authors,
In this round of revision, you strongly have focused your efforts on the points unveiled by different reviewers. You have improved your manuscript for journal style and clarity. In this state, the manuscript could be suitable for publication.
Reviewer 4 Report
The authors adequately clarified the points raised